# A New Perspective on Using Glycols in Glutamate Biosensor Design: From Stabilizing Agents to a New Containment Net

**Andrea Bacciu [1], Paola Arrigo [1], Giovanna Delogu [2], Salvatore Marceddu [3], Patrizia Monti [4], Gaia Rocchitta [1,5,*] and Pier Andrea Serra [1,3,5]**

[1] Dipartimento di Scienze Mediche, Chirurgiche e Sperimentali, Università degli Studi di Sassari, Viale San Pietro 43/b, 07100 Sassari, Italy; andreabacciu90@gmail.com (A.B.); pa1989@live.it (P.A.); paserra@uniss.it (P.A.S.)

[2] Istituto CNR di Chimica Biomolecolare-UOS Sassari, Traversa La Crucca 3, I-07100 Sassari, Italy; giovanna.delogu@icb.cnr.it

[3] Istituto CNR di Scienze delle Produzioni Alimentari-UOS Sassari, Traversa La Crucca 3, 07100 Sassari, Italy; salvatore.marceddu@ispa.cnr.it

[4] Dipartimento di Agraria and Unità di Ricerca Istituto Nazionale di Biostrutture e Biosistemi, Università degli Studi di Sassari, Viale Italia 39, 07100 Sassari, Italy; patrizia.bth.ss@gmail.com

[5] Mediterranean Center for Disease Control, University of Sassari, 07100 Sassari, Italy

* Correspondence: grocchitta@uniss.it

**Abstract:** Glutamate is a major excitatory neurotransmitter in the brain. It is involved in many normal physiological brain activities, but also neurological disorders and excitotoxicity. Hence, glutamate measurement is important both in clinical and pre-clinical studies. Pre-clinical studies often use amperometric biosensors due to their low invasiveness and the relatively small size of the devices. These devices also provide fast, real-time measurements because of their high sensitivity. In the present study, diethylene glycol (DEG), neopentyl glycol (NPG), triethylene glycol (TEG), and glycerol (GLY) were used to increase the long-term stability of glutamate biosensors. The evaluation was made by measuring variations of the main enzymatic ($V_{MAX}$ and $K_M$) and analytical (Linear Region Slope (LRS)) parameters. Of the glycols tested, TEG was the most promising stabilizer, showing about twice as high $V_{MAX}$ maintained over a greater duration than with other stabilizers tested. It is also yielded the most stable linear region slope (LRS) values over the study duration. Moreover, we highlighted the ability of glycols to interact with enzyme molecules to form a containment network, able to maintain all the layered components of the biosensor adhering to the transducer.

**Keywords:** glutamate biosensor; glycols; over-time stability; containment net

---

## 1. Introduction

Glutamate is a major excitatory neurotransmitter in the brain [1,2] implicated in normal physiological brain activities [3,4]. These activities include neurotransmission, learning and memory processes, synaptic plasticity, neuronal development, and aging [5,6]. However, glutamate is also implicated in dysfunction, with high levels of glutamate associated with neuronal death [7] and neurological disorders [8–10]. Glutamate is involved in excitotoxicity as well, due to an over-activation of glutamate receptors [11,12]. Given the importance of glutamate in both normal and abnormal brain function, the measurement of glutamate levels is relevant in both clinical and pre-clinical studies.

In clinical investigations, various techniques have been used to measure glutamate, including spectroscopic and neuroimaging techniques [13,14] used to correlate glutamate levels to brain

dysfunctions. As previously reported [15], these techniques are particularly suitable for patients, as well as animal models, for their non-invasiveness. However, they are characterized by low spatial and temporal resolution, as well as a moderately high limit of detection [16].

In some studies, glutamate concentrations were measured in blood samples using high-performance liquid chromatography (HPLC) [17,18]. While blood sampling is more invasive than imaging, the approach is still suitable for use in patients, and HPLC is reliable and sensitive enough to measure glutamate concentrations in blood samples, as well as in brain samples [17,19,20].

The pathophysiological role of glutamate in the CNS and etiology of some glutamate-related diseases is poorly understood. Therefore, the detection of glutamate in pre-clinical studies, in animal models, has become increasingly important.

Since the 1990s, microdialysis has been widely used for monitoring neurochemicals in the CNS, including glutamate. It is a technique where a probe is inserted into tissue. Moreover, the coupling with HPLC allowed the measurements of numerous compounds present in microsdialysates obtained from extracellular fluids [21–23]. Low temporal resolution, the relatively large probe size (~200 μm), as well as the need for coupling with an analytical technique to quantify analytes in the microdialysate, are the fundamental limitations [16,24].

In recent decades, the advent of biosensors capable of monitoring neurochemicals has advanced the field. Biosensors are "self-standing devices," that both sample and monitor the analytes of interest. Amperometric biosensors are suitable for measurements of neurochemicals. Mainly because they are smaller and hence less invasive, and they also perform fast, real-time measurements with high sensitivity. It has been demonstrated that glutamate concentrations in the brain are known to be comprised between 1 and 10 μM, depending on the brain region [24,25]. There are also some pieces of evidence that basal glutamate can be found in the submicromolar range due to the high efficiency of glutamate transporters [26,27].

Glutamate concentrations in the brain are low, so other analytes present in the brain can interfere with detection. Hence, to improve biosensor performance, shielding from interfering species is important [24], particularly when biosensors are implanted in animal models, both for short and medium durations. One of the most prevalent interfering species is ascorbic acid (AA), which is present at considerably higher concentrations (~500 μM) than glutamate in extracellular fluids (ECF) [24,25,28]. In addition, to reduce the effect of interfering species, sensitivity, efficiency, and performance over time are all areas where improvements can be made in biosensor design.

In the present study, different glutamate biosensor designs were designed and tested. All designs employed L-glutamate oxidase (GluOx), which selectively catalyzes the conversion of L-glutamate to produce an electrochemical signal, as follows:

$$\text{L-Glutamate} + \text{H}_2\text{O} + \text{GluOx/FAD} \rightarrow \alpha\text{-ketoglutarate} + \text{NH}_3 + \text{GluOx/FADH}_2 \qquad (1)$$

$$\text{GluOx/FADH}_2 + \text{O}_2 \rightarrow \text{GluOx/FAD} + \text{H}_2\text{O}_2 \qquad (2)$$

$$\text{H}_2\text{O}_2 \xrightarrow{\text{Pt}} \text{O}_2 + 2\text{H}^+ + 2\text{e}^- \qquad (3)$$

Hydrogen peroxide (H$_2$O$_2$), which is generated during the enzymatic reaction, is measured on a platinum (Pt) surface when an anodic potential of +0.7 V vs. Ag/AgCl is applied [29–32]. The current generated from H$_2$O$_2$ oxidation is proportional to the glutamate concentration in the matrix.

As previously published [29–31,33–35], interfering species can be successfully excluded from the electrode surface by electrodeposition of a polyorthophenylenediamine (PPD) polymer.

Sensitivity, efficiency, and performance over time are crucial and continually improving with new biosensor designs [24,35,36]. The most common approach to improve the efficiency of biosensors is the use of polyethyleneimine (PEI) [24,25,28–30,37]. Very recently, it has been demonstrated that glycols (and polyols in general) can improve biosensor efficiency in the presence of PEI, as well as increase stability over time [24,38]. Indeed, it has been shown that propylene glycol (PG) improves

glutamate biosensor activity over time [24]. Some polyols can act as enzyme stabilizers by interacting with proteins to control the enzyme microenvironment and protect against denaturation [24,39].

Different parameters were considered to evaluate the stability of the glutamate biosensors over time. Among the most important evaluated was $V_{MAX}$, which represents the maximum enzymatic rate at which substrate is converted into end products, when all catalytic sites are saturated, and enzymes can form an enzyme-substrate complex [24,35,38,40,41]. $V_{MAX}$ indicates the number of active enzyme molecules loaded on the biosensor surface [24,34,35,37,38,42].

In addition, the Michaelis–Menten constant, $K_M$, was determined as this parameter is correlated with the affinity between the enzyme and substrate, giving the substrate concentration that produces half of the $V_{MAX}$ [25,34,35]. $K_M$ variations have been linked to the linear operating range (equal to about a half of $K_M$). Indeed, some authors have demonstrated that a high value of $K_M$ produces a broad linear range for biosensors [30,35,43]. Moreover, $K_M$ is essential for determining one of the most critical analytical parameters: the linear region slope (LRS). LRS indicates the sensitivity of the biosensor for the substrate and can be considered a $K_M$ function [35,43,44].

In this study, different glycols were examined for their ability to improve the sensitivity and stability over time of glutamate biosensors. The compounds were chosen based on their different structures and possible interactions with GluOx.

## 2. Materials and Methods

### 2.1. Chemicals and Reagents

All compounds were bought from Sigma-Aldrich (Milan, Italy) and used as supplied. Phosphate-buffered saline (PBS, 0.05 M) solutions used for calibrations and electropolymerizations were prepared using 0.15 M NaCl, 0.05 M $NaH_2PO_4$ and 0.04 M NaOH, and adjusted to pH 7.4. The enzyme stabilizer polyethyleneimine (PEI, 1%) was obtained by dilution of a stock solution (50% w/v) in bidistilled water. Neopentyl glycol (NPG, 0.1%), triethylene glycol (TEG, 0.1%), diethylene glycol (DEG, 0.1%), and glycerol (GLY, 0.1%) solutions were prepared in double-distilled water. o-Phenylenediamine monomer (OPD, 0.3 M) solutions were made by dissolving monomer powder in deoxygenated PBS. Glutamate Oxidase (GluOx, 400 U/mL in PBS) was from Yamasa Corp. Stock solutions of $H_2O_2$ (0.1 M) were prepared in double-distilled water. Sodium glutamate (Glut, 1 M and 0.01 M) solutions were prepared in double-distilled water. Ultrapure (>99.9%) nitrogen ($N_2$) was purchased from Sapio s.r.l Special Gases Division (Caponago, Italy). Teflon® insulated Platinum/Iridium wire (Pt-Ir, 90:10, Ø = 125 μm) was acquired from Advent Research Materials (Eynsham, UK).

### 2.2. Instrumentation and Software

All electrochemical experiments were performed using a classical three-electrode cell. It consisted of a beaker holding 20 mL of PBS, four glutamate biosensors as working electrodes, an Ag/AgCl (NaCl 3M) electrode (Bioanalytical Systems, Inc., West Lafayette, IN, USA), and a stainless needle as the auxiliary electrode. A four-channel potentiostat (eDAQ Quadstat, e- Corder 410, eDAQ Europe, Poland) and the software Chart (v 5.5, eDAQ Europe, Poland) were employed for all electrochemical procedures.

### 2.3. Biosensor Construction and Characterization

All biosensors were constructed using the same cylindrical geometry (1 mm in length and 125 μm in diameter) (Figure 1). Briefly, a 3 cm portion of Pt/Ir wire was cut, and from one edge, 3 mm of Teflon® insulation was removed to allow the bare metal to be welded to a support. From the other edge of the wire, 1 mm of bare metal was exposed for modifications. On Day 0, ortho-phenylenediamine polymer (PPD) electropolymerization was carried out by immersing the Pt/Ir wires in an OPD 0.30 M solution prepared in deoxygenated PBS ($N_2$ 100%). It was prepared by bubbling ultrapure $N_2$ gas in 12 mL of PBS for 30 min, and by applying a positive potential of +0.7 V vs. Ag/AgCl electrode (RE) for 30 min. After polymerization, Pt/p-OPD cylinders were rinsed in pure water and dipped twice in a PEI

1% solution, waiting 5 min between each dipping. At that point, electrodes were dipped in the GlutOx solution (400 U/mL) for five times, leaving to dry for 5 min at room temperature between each dip. Finally, one layer of 0.1% glycol was deposited on the top of each biosensor using a single dip, after which the biosensors were left to dry at room temperature for 30 min. Depending on the glycol used, different biosensors were manufactured with the following general design:

$$Pt_c/PPD/PEI(1\%)_2/GlutOx_5/Glycol(0.1\%)$$

where Glycol is represented by DEG, NPG, TEG, and GLY.

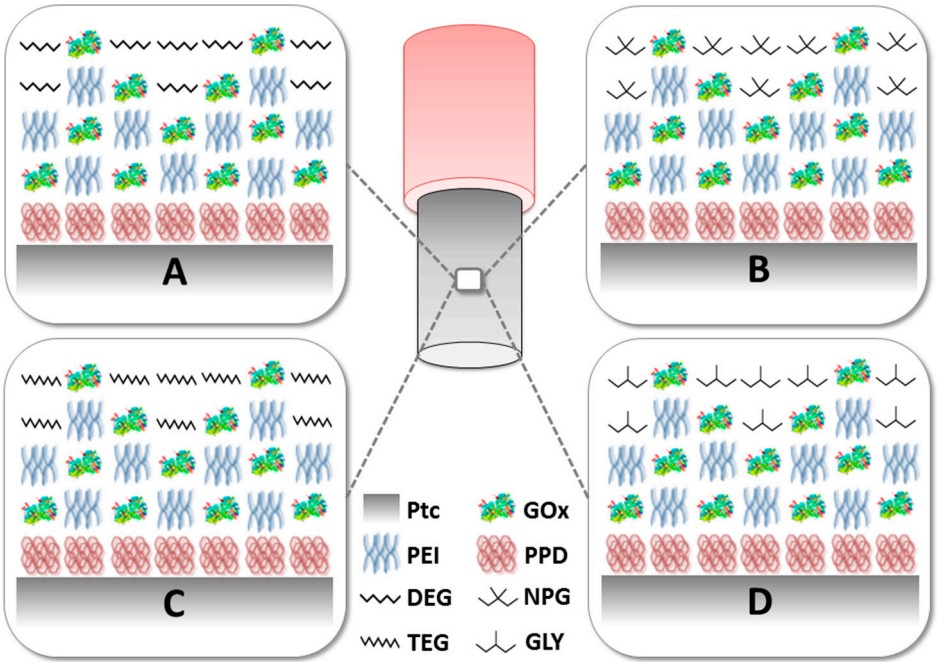

**Figure 1.** Schematic representation of the glutamate biosensors described in this study **Panel A**: $Pt_c/PPD/PEI(1\%)_2/GlutOx_5/DEG(0.1\%)$; **Panel B**: $Pt_c/PPD/PEI(1\%)_2/GlutOx_5/NPG(0.1\%)$; **Panel C**: $Pt_c/PPD/PEI(1\%)_2/GlutOx_5/TEG(0.1\%)$; **Panel D**: $Pt_c/PPD/PEI(1\%)_2/GlutOx_5/GLY(0.1\%)$. $Pt_c$: Pt cylinder 1 mm long; GlutOx: L-glutamate oxidase; PPD: ortho-phenylenediamine polymer; PEI: polyethyleneimine, DEG: diethylene glycol, NPG: neopentyl glycol, TEG: triethylen glycol, GLY: glycerol, PG: propylene glycol. The subscript number represents the number of dips and in brackets, the concentration of the glycol.

Moreover, a control biosensor design was constructed, where the final coating was omitted:

$$Pt_c/PPD/PEI(1\%)_2/GlutOx_5$$

Biosensors were put in 20 mL of fresh PBS and then polarized at a constant potential of +0.7 V vs. Ag/AgCl overnight for stabilization. On Day 1, a full glutamate calibration ranging from 0 to 50 mM was conducted in fresh PBS by the addition of fixed volumes of Glut stock solutions (0.01 M and 1 M) (Figure 2). In Figure S1 of the Supplementary Material, calibration plots are shown, from 21 days, of the two representative glycol designs (DEG and TEG). All designs were subjected to the same calibration protocol, which was repeated on Day 7, 14, and 21 to assess biosensor aging. Biosensor performances were evaluated from calibration data in terms of enzymatic kinetic ($V_{MAX}$ and $K_M$) and analytical performances (linear region slope—LRS and LOD). After each calibration, all biosensors were rinsed in double-distilled water and stored at +4 °C.

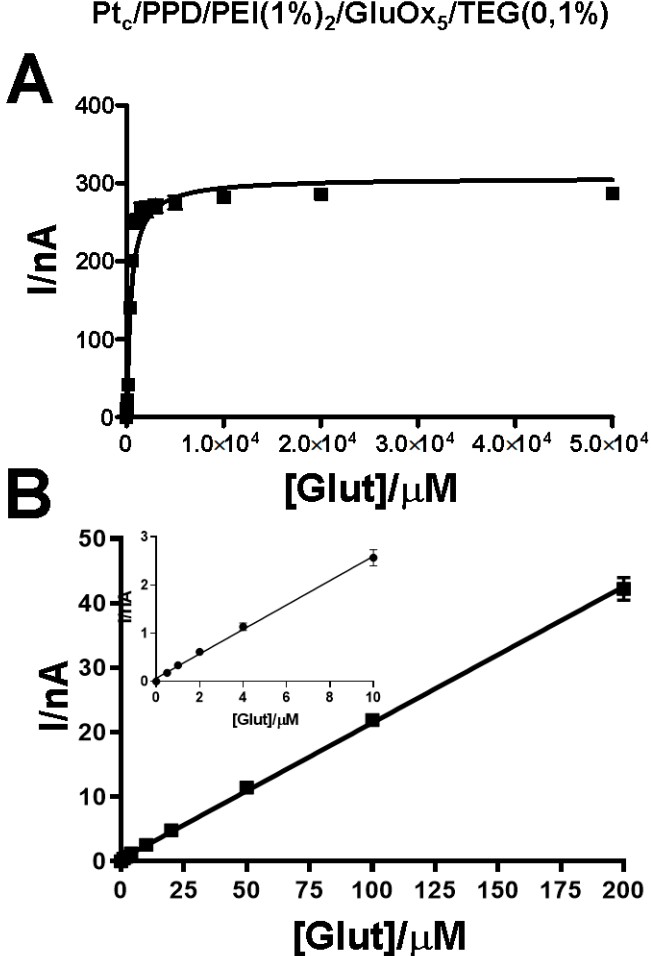

**Figure 2.** Representative plot of biosensor loading TEG (0.1%) calibrations. The non-linear fitting of data (**Panel A**) was performed in the range 0–5 × 10⁴ μM of concentrations of glutamate. Meanwhile, the linear fitting of data (**Panel B**) was made in the 0–200 μM range In the inset of **Panel B**, linear regression data obtained from 0–10 μM range of concentrations of glutamate is highlighted. Data were obtained from Day 1 calibration.

All the electrochemical procedures as electropolymerization or calibrations were carried out using constant potential amperometry (CPA).

### 2.4. Statistical Analysis

After in vitro calibrations, biosensor currents were plotted versus Glut concentrations. Linear regression was calculated at low Glut concentrations (0–0.2 mM), to determine the value of LRS, while a nonlinear fitting (Michaelis–Menten equation) was performed on the entire concentration range (0–50 mM) to evaluate $V_{MAX}$ and $K_M$. While $V_{MAX}$ was expressed as nA, the LRS was expressed as current density (nA μM$^{-1}$ cm$^{-2}$). Statistical significance (p values) was evaluated by ANOVA using GraphPad Prism v 5.02 software.

As previously published [29], the limit of detection (LOD) was determined using a statistical method based on the standard deviation (σ) of the response and the LRS of the calibration curve.

### 2.5. Scanning Electron Microscopy

Biosensors were also examined by scanning electron microscopy (SEM). After construction, biosensors with different glycols were immobilized on a holder for gold coating. As previously published [38], after plating, microphotographs were collected using a DSM 962 Zeiss conventional

SEM (Oberkochen, Germany) by applying an accelerating voltage of 25 kV. Microphotographs were taken at different magnifications of 2000, 5000, and 1000×.

## 3. Results

As shown in Figure 2, kinetic parameters ($V_{MAX}$ and $K_M$) were calculated using nonlinear fitting of calibrations data in the range of 0–50 µM of glutamate (Panel A). Meanwhile, the analytical parameter LRS was obtained by means of linear fitting of calibration data in the range between 0 and 200 µM of glutamate (Panel B). In the inset of Panel B, the graph inherent in the concentration range 0–10 µM of glutamate is highlighted.

### 3.1. Impact of Glycols on $V_{MAX}$ Over Time

Figure 3 shows the effect of the absence of glycol as a final layer or the presence of different glycols on the $V_{MAX}$ over 21 days. The control design at Day 1 showed a $V_{MAX}$ equal to 148.60 ± 3.48 nA, showing a significant decrease in $V_{max}$ starting from Day 7 (126.60 ± 3.18 nA) and further decreasing up to Day 21 (13.36 ± 1.37 nA).

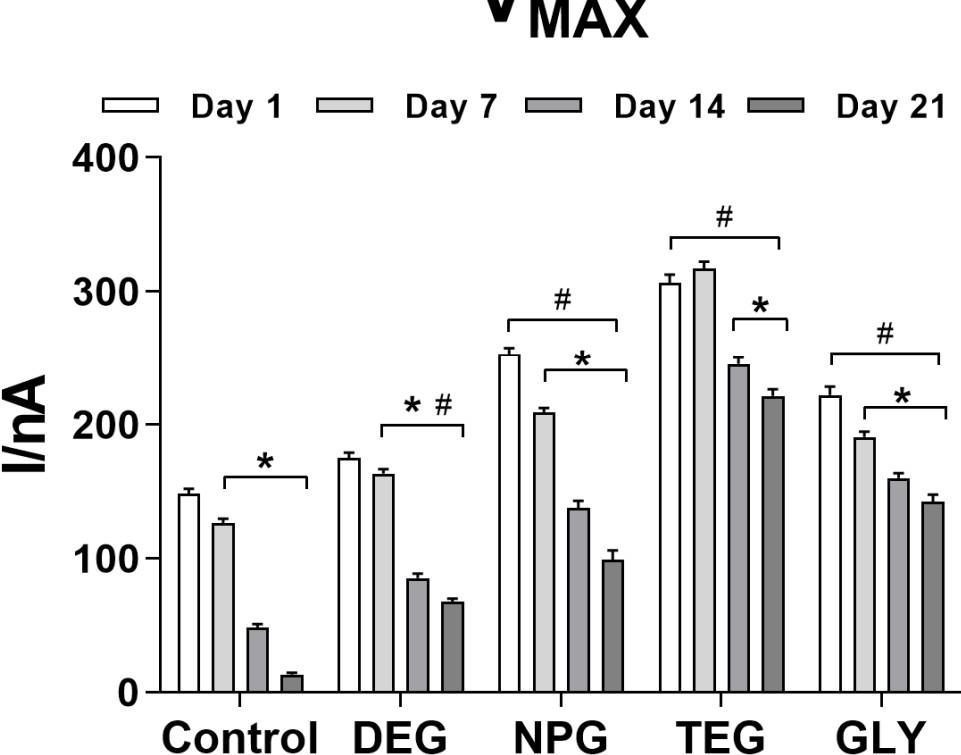

**Figure 3.** Bar plot of the variation of $V_{MAX}$ for different biosensor designs (n = 4): $Pt_c$/PPD/PEI(1%)$_2$/GlutOx$_5$/Glycol(0.1%). $Pt_c$: Pt cylinder 1 mm long; GlutOx: L-glutamate oxidase; PPD: ortho-phenylenediamine polymer; PEI: polyethyleneimine, DEG: diethylene glycol, NPG: neopentyl glycol, TEG: triethylen glycol, GLY: glycerol. The subscript number represents the number of dips and in brackets, the concentration of the glycol. Values are expressed as mean ± SEM. * $p < 0.001$ vs. Day 1 of the respective group. # $p < 0.001$ vs. control design.

The presence of DEG displayed a $V_{MAX}$ of 175.30 ± 3.78 nA on Day 1, while a small but significant decrease ($p < 0.001$ vs. Day 1) was observed on Day 7 (163.50 ± 3.27 nA) with further reductions on Days 14 and 21, dropping to 67.71 ± 2.45 nA.

A comparable trend was observed when NPG was used. A reasonably high $V_{max}$ was obtained on Day 1 (253.20 ± 4.53 nA), but a significant decrease ($p < 0.001$ vs. Day 1) was observed over time, reaching a minimum value of 99.16 ± 4.53 nA at Day 21.

TEG biosensors produced the highest values of $V_{MAX}$ between Day 1 and 7 when the parameter was equal to 317.10 ± 5.46 nA (not statistically different from Day 1). Over time, $V_{MAX}$ decreased moderately but significantly ($p < 0.001$ vs. Day 1) on both Days 14 and 21, to a minimum of 221.50 ± 4.98 nA.

GLY-loaded biosensors generated a $V_{MAX}$ of 222.20 ± 6.21 nA at Day 1, and a constant significant decrease ($p < 0.001$ vs. Day 1) was observed over time to 142.50 ± 5.22 nA at Day 21.

All glycol-containing designs showed a significant difference compared to the glycol-free design in terms of $V_{MAX}$ ($p < 0.001$), over 21 days.

### 3.2. Impact of glycols on $K_M$ over time

Figure 4 shows the effect of the presence or the absence of glycol as a final layer on the $K_M$ over 21 days. All glycols produced an average value of $K_M$ of about 450 μM on Day 1, compared to about 600 μM for the control design. In all designs, a general increase in $K_M$ was observed over time. In particular, the absence of glycol as a final layer produced a significant improvement over time, with the $K_M$ reaching about 1490 μM. This design showed the highest $K_M$ values among all the considered designs.

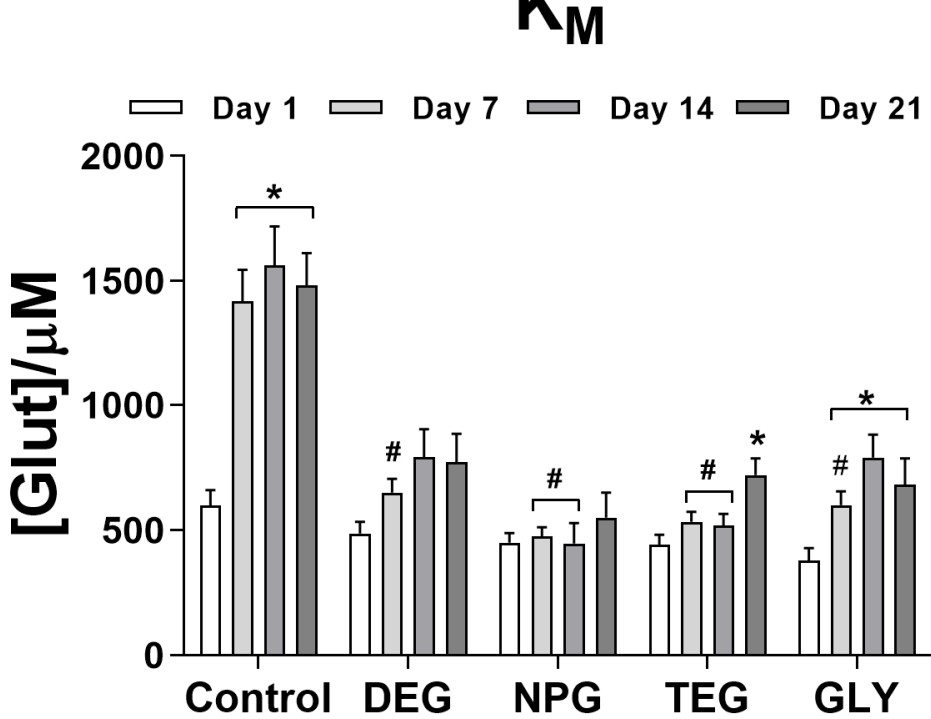

**Figure 4.** Bar plot of the variation of $K_M$ for different biosensor designs (n = 4): $Pt_c$/PPD/PEI(1%)$_2$/GlutOx$_5$/Glycol(0.1%). $Pt_c$: Pt cylinder 1 mm long; GlutOx: L-glutamate oxidase; PPD: ortho-phenylenediamine polymer; PEI: polyethyleneimine, DEG: diethylene glycol, NPG: neopentyl glycol, TEG: triethylen glycol, GLY: glycerol. The subscript number represents the number of dips and in brackets, the concentration of the glycol. Values are expressed as mean ± SEM. * $p < 0.001$ vs. Day 1 of the respective group. # $p < 0.001$ vs. control design.

$K_M$ significantly increased from Day 7 up to Day 21 for the DEG design, reaching a final value of 775.50 ± 113.80 μM. Unlike all other glycols, NPG sensors did not exhibit any statistically significant variation of $K_M$ over 21 days. TEG produced the general stability of $K_M$ until Day 14, reaching about

550 μM. On Day 21, the $K_M$ increased significantly up to 720.60 ± 66.96 μM. Biosensors with GLY showed significant increases in $K_M$ over time, reaching a maximum of 790.10 ± 92.57 μM on Day 14.

### 3.3. Impact of Glycols on Analytical Parameters Over Time

In Figure 5, the variation of LRS over time of different biosensor designs is shown and expressed as a function of current density. The control design showed the lowest values of LRS, from Day 1 (36.346 ± 0.487 nA μM$^{-1}$ cm$^{-2}$). Moreover, the absence of glycol resulted in a significant decrease, starting from Day 7 (6.121 ± 0.268 nA μM$^{-1}$ cm$^{-2}$) to Day 21 (2.366 ± 0.244 nA μM$^{-1}$ cm$^{-2}$).

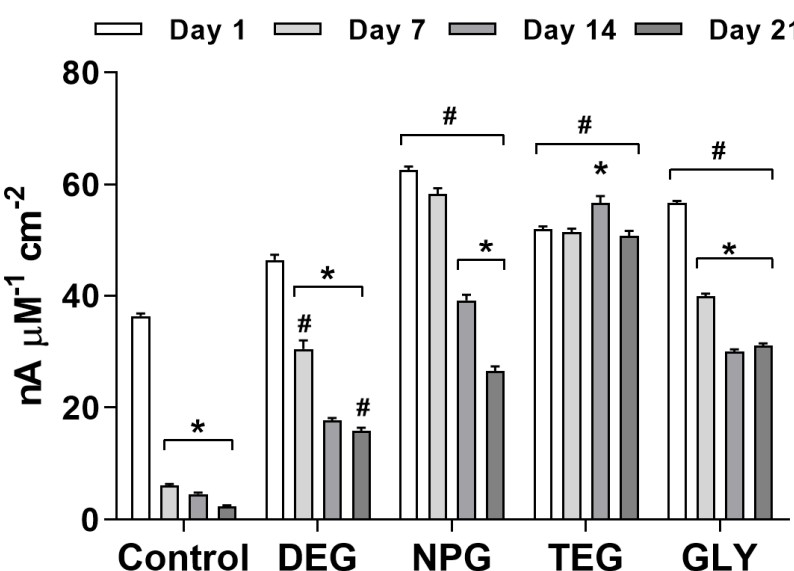

**Figure 5.** Bar plot of the variation of LRS for different biosensor designs (n = 4): Pt$_c$/PPD/PEI(1%)$_2$/GlutOx$_5$/Glycol(0.1%). Pt$_c$: Pt cylinder 1 mm long; GlutOx: L-glutamate oxidase; PPD: ortho-phenylenediamine polymer; PEI: polyethyleneimine, DEG: diethylene glycol, NPG: neopentyl glycol, TEG: triethylen glycol, GLY: glycerol. The subscript number represents the number of dips and in brackets, the concentration of the glycol. Values are expressed as mean ± SEM. * $p < 0.001$ vs. Day 1 of the respective group. # $p < 0.001$ vs. control design.

The design with DEG showed a substantial, significant ($p < 0.001$ vs. Day 1) decrease in LRS from Day 7 (30.469± 1.597nA μM$^{-1}$ cm$^{-2}$) and further reductions on Days 14 and 21 (4.583 ± 0.279 nA μM$^{-1}$ cm$^{-2}$ and 2.366 ± 0.224 nA μM$^{-1}$ cm$^{-2}$, respectively).

The design with NPG produced the highest LRS value on Day 1 (62.593 ± 0.608 nA μM$^{-1}$ cm$^{-2}$). On Day 7, the LRS decreased but not significantly, while at Day 14 and Day 21 the decreases were significant (39.160 ± 1.016 nA μM$^{-1}$ cm$^{-2}$ and 26.543 ± 0.822 nA μM$^{-1}$ cm$^{-2}$, respectively).

The design with TEG showed an interesting trend in LRS over time, different from all other designs. While this design did not yield the highest LRS values, LRS was constant over time, with an average value of about 51.850 nA μM$^{-1}$ cm$^{-2}$. A significant difference was found for Day 14 (56.790 ± 1.160 nA μM$^{-1}$ cm$^{-2}$; $p < 0.001$ vs. Day 1).

The use of GLY in the biosensor design yielded a decrease in LRS, similar to DEG and NPG. This design showed a Day 1 value of 56.691 ± 0.437 nA μM$^{-1}$ cm$^{-2}$, with a decrease to 39.877 ± 0.464 nA μM$^{-1}$ cm$^{-2}$ by Day 7, significantly lower ($p < 0.001$) than Day 1. LRS further decreased significantly ($p < 0.001$ vs. Day 1) on Days 14 and 21 (30.025 ± 0.421 nA μM$^{-1}$ cm$^{-2}$ and 031.086 ± 0.424 nA μM$^{-1}$ cm$^{-2}$, respectively).

Comparisons between the different designs demonstrated a significant difference compared to the glycol-free design in terms of $V_{MAX}$ ($p < 0.001$).

As shown in Figure 6, and as expected, the control biosensor without any glycol had the best LOD on Day 1 ($0.164 \pm 0.052$ μM), but significant decreases over time from Day 7 ($p < 0.001$ vs. Day 1), with the LOD increasing to $3.256 \pm 1.053$ μM by Day 21.

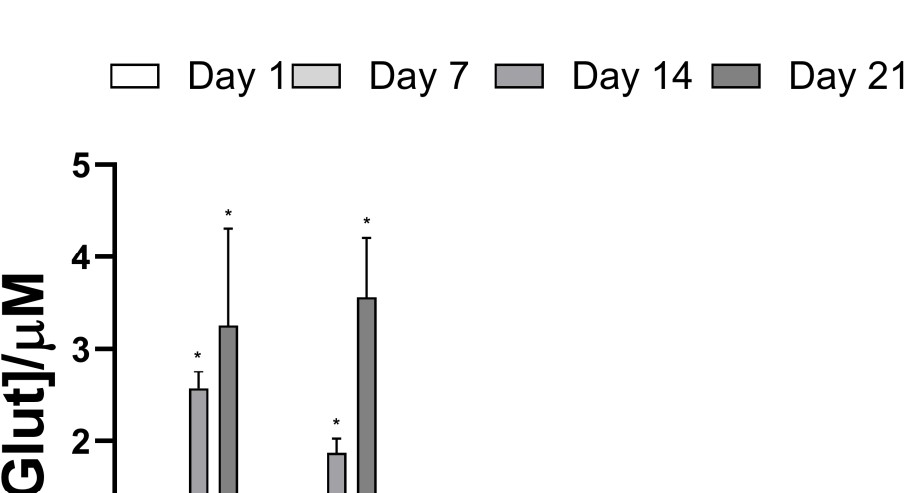

**Figure 6.** Bar plot of the variation of LOD for different biosensor designs (n = 4): $Pt_c/PPD/PEI(1\%)_2/GlutOx_5/Glycol(0.1\%)$. $Pt_c$: Pt cylinder 1 mm long; GlutOx: L-glutamate oxidase; PPD: ortho-phenylenediamine polymer; PEI: polyethyleneimine, DEG: diethylene glycol, NPG: neopentyl glycol, TEG: triethylen glycol, GLY: glycerol. The subscript number represents the number of dips and in brackets, the concentration of the glycol. Values are expressed as mean ± SEM. * $p < 0.001$ vs. Day 1 of the respective group.

Designs with glycols as the final layer had higher LODs on Day 1 (average value = 0.450 μM). However, all but the DEG design had better LODs over time than the control design.

For the NPG design, the LOD was $0.780 \pm 0.042$ μM and $0.915 \pm 0.092$ μM on Days 14 and 21, respectively. A slight decrease relative to Day 1, although not significant, was recorded on Day 7.

With regard to TEG and GLY, there were no significant differences in the LOD values over the observation period, with LODs of about 0.400 μM and 0.497 μM, respectively.

### 3.4. Scanning Electron Microscopy (SEM) Analysis

Following previously published studies [38,45,46], the designs were further characterized by SEM to examine surface characteristics. Microphotographs were taken with a conventional SEM, at Day 1 after construction, with a magnification of 2000×, as shown in Figure 7. Microphotographs with the magnifications of 5000 and 1000× are reported in Figure S2 of the Supplementary Material.

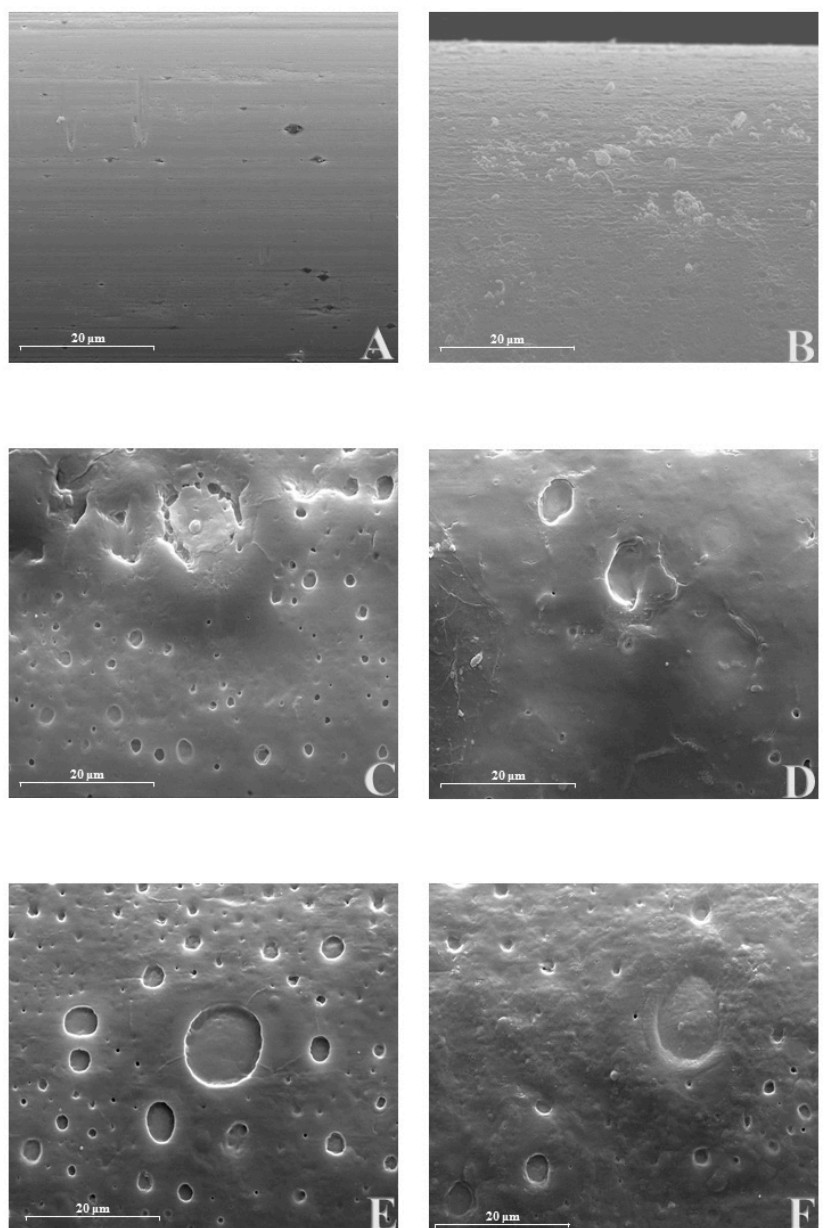

**Figure 7.** Scanning electron microscopy (SEM) of different biosensor designs at 2000x magnification. **Panel A**: $Pt_c$; **Panel B**: $Pt_c$/PPD; **Panel C**: $Pt_c$/PPD/PEI$(1\%)_2$/GlutOx$_5$/DEG(0.1%); **Panel D**: $Pt_c$/PPD/PEI$(1\%)_2$/GlutOx$_5$/NPG(0.1%); **Panel E**: $Pt_c$/PPD/PEI$(1\%)_2$/GlutOx$_5$/TEG(0.1%); **Panel F**: $Pt_c$/PPD/PEI$(1\%)_2$/GlutOx$_5$/GLY(0.1%).

As shown in Figure 6, all the biosensor's surfaces differed from bare Pt (Panel A) because of the presence of different layered components. In panel B, surface modification due to the presence of PPD was evident, which was consistent with previous findings [38]. In panels C, D, E, and F, the deposition of PEI, GlutOx, and glycols resulted in the formation of craters on the surface, with a peculiar size and distribution for each design. DEG and NPG (Panel C and D of Figure 7, respectively) showed craters on average that were smaller than TEG and GLY (Panel E and F of Figure 7, respectively). A closer look at these cavities (Figure S2) suggested a degree of depth greater for DEG (Panels 1A and 1B from Figure S2) than all the others (Panels 2A-B, Panels 3A-B, Panels 4A-B of Figure S2). Moreover, DEG (Panel C), together with TEG (Panel E) had better-sharpened edges compared to NPG (Panel D) and GLY (Panel F).

## 4. Discussion

In the present study, the impact of different glycols on glutamate biosensor activity and stability over time was studied to develop better biosensors for in vivo applications. As previously described [24,38], glycols, and polyols in general, stabilize enzyme activity by forming a stable hydration layer around the enzyme molecules, which maintains the native enzyme conformation (folded state) [38,47,48] and diminishes unfavorable interactions between the enzyme and surrounding solution [38,49]. This is important for biosensor stability as enzyme activity is the first parameter to suffer from prolonged biosensor use and enzyme denaturation during immobilization for biosensor manufacture [34,50]. The loss of enzyme activity could also be due to a rearrangement over time within the biosensor, which could result in less favorable interactions with the substrate over time, as the enzymes are not covalently linked to the biosensor platinum surface [38]. $V_{MAX}$ reflects the number of active molecules present on the transducer surface [24,34,35,37,38,42], so an increase (or a decrease) in $V_{MAX}$ reflects a variation in enzyme activity.

In our experiments, a control design was investigated, which lacked glycol as the final layer. As expected, in this design, the $V_{MAX}$ decreased from Day 7, demonstrating that the lack of a containment network did not prevent the loss of enzymatic activity, causing a sudden loss of the biosensor performances.

In this study, all the glycol designs showed better $V_{MAX}$ than the control. While DEG and NPG showed a consistent loss of enzymatic activity (with a sustained decrease in $V_{MAX}$), TEG and GLY showed a lesser decrease. In particular, TEG showed a slight (although not significant) increase between Day 1 and 7 and the smallest pronounced decrease among all designs. The obtained $V_{MAX}$ values are in line with measurements of a comparable factor $J_{max}$ in the literature [33], where a conversion could easily be obtained considering the surface area of the 1 mm Pt/Ir cylinder ($4.05 \times 10^{-3}$ cm$^2$). These data confirmed that glycols could preserve enzymatic activity, in particular, TEG. Notably, all glycol compounds tested were monomeric and easier to handle than PG [38]. Their deposition increased the retention of enzyme molecules on the biosensor surface.

Usually, containment net or crosslinking agents are used to entrap layered components on glutamate biosensor surfaces [24,25,37,51]. Aggressive crosslinking agents are sometimes employed, such as glutaraldehyde, that can affect biosensor enzyme performance over time. Very recently [24], it has been found that PG, a small monomeric molecule never used before in biosensing, exhibits the same characteristics as the polyethylene glycol diglycidyl ether (PEDGE). This has been widely used as a cross-linking agent for biosensors fabrication [24,25]. In the present study, some of the glycols tested showed a behavior similar to PG. Their deposition plausibly created a containment net, likely due to the intramolecular glycol–protein interactions, able to maintain the layered compound on the transducer surface, improving enzyme performance.

In the past, glycerol has been shown to stabilize proteins by influencing the hydration environment, as well as establishing hydrogen bonds due to interaction of the outer oxygen atoms of glycerol with nitrogen and oxygen atoms in the amino acids composing proteins [52]. The study identified glycerol–protein interactions even at the hydrophobic surface of proteins. These interactions are more pronounced with less hydrophilic glycerol. The occurrence of hydrogen bonds, which are characterized by weak to moderate stabilization energies [53] aside from hydrophobic interactions, is likely responsible for the formation of a glycol–protein network. This network can maintain biosensor components on the transducer surface.

Among the glycols used here, GLY and TEG had the most favorable impact on $V_{MAX}$ (highest values) while also exhibiting the smallest decrease over time. In particular, for TEG, after an initial (although not significant increase) on Day 7, $V_{MAX}$ values remained relatively high to Day 21, indicating good enzymatic activity. It has recently been shown that TEG can establish hydrogen bonds not only with terminal OH groups but also with ether oxygens atoms [54], resulting in higher enzyme stability over time. Furthermore, the structural flexibility of TEG, and likely, the molecular length allows this compound to be positioned between the enzyme amino acids to maximize interactions, including

simultaneous hydrophilic (such as hydrogen bonds) and hydrophobic interactions. Such interactions could result in high and persistent values of $V_{MAX}$ over time.

Concerning $K_M$, the deposition of glycols influenced the affinity of the enzyme for the substrate. In general, an increase over time was observed, as expected. Surprisingly, NPG and TEG showed relative stability in $K_M$ values over 14 days, more likely due to stronger intramolecular forces established by these glycols with the enzyme compared to the other glycols. Once again, the absence of a containment net led to higher $K_M$ values than in the control design, and therefore, to worse performance compared to designs with glycols.

As previously noted [35,43,44], LRS is one of the essential analytical parameters. It reflects biosensor efficiency in converting the substrate into byproducts but also represents sensitivity to the analyte. LRS is a function of $V_{MAX}$ and $K_M$ [43], so these two parameters affect the LRS.

As expected, and highlighted above, the absence of a containment network resulted in a loss of active enzymes on the transducer surface. This phenomenon had an important impact on LRS values. In fact, in the control design in which glycol was absent, from Day 7 and up to Day 21, there was a significant decrease in LRS. Almost all of the glycols used here yielded interesting LRS values, except for DEG, which revealed quite low values and was particularly unstable over time. Other glycols yielded quite high LRS values, which decreased with time, except for TEG, which surprisingly produced stable LRS values over the study period. The LRS values were found to be in line with the values already present in the literature [25].

Glycols had an interesting effect on the LOD values. All glycols increased LOD values on Day 1 when compared to the glycol-free design. Biosensors without glycol exhibited increasing LODs starting from Day 7 until Day 21. Among the glycols, only DEG and NPG showed a comparable behavior to the glycol-free design. At the same time, TEG and GLY gave LOD values that did not significantly change over the study period, demonstrating good analytical performance.

The behavior of biosensors with DEG over time is probably due to the few, and not very effective, interactions between glycol and the enzyme. This indicates that hydrophobicity and an appropriate molecular length glycol can positively influence biosensor performance. DEG–protein interactions did not allow protection of enzyme molecules, nor their effective orientation in space, reducing the enzyme-substrate interactions, as denoted by the low values of $V_{MAX}$ and LRS, and the high values of $K_M$. Moreover, DEG did not have adequate stabilizing effects.

Biosensors with NPG and GLY showed comparable behavior over time. The relatively higher values of $V_{MAX}$ and LRS indicated enzyme stabilization, suggesting efficacious intermolecular interactions, maintaining a higher number of active molecules on the transducer surface. These phenomena may be attributed to the ability of the glycols to establish effective hydrogen bonds through the OH moieties, but also to create hydrophobic connections, the latter being more pronounced in NPG.

Surprisingly, the deposition of TEG revealed interesting results in terms of enzymatic kinetics ($V_{MAX}$ and $K_M$) and analytical efficacy (LRS and LOD) over time. The high number of active molecules and high sensitivity, as well as the constant enzyme affinity, reflect the elevated capability of TEG to stabilize enzymatic activity, over the time frame studied. Moreover, the persistence of many active enzyme molecules that were well-oriented for interaction with the substrate could be due to hydrophilic and hydrophobic interactions of TEG through OH moieties and ether oxygens [55,56]. Moreover, the structural flexibility of TEG probably allows better interactions with the amino acids of the enzyme.

High vacuum conditions used for SEM imaging strongly influenced the presence of water in the sample, thereby altering hydrophilic interactions. Since glycols form the external layer, enzyme and PEI are mediated by water molecules, and $H_2O$ removal can lead to a collapse of the network. In this way, craters observed on biosensor surfaces would represent the tears of the texture. On the other hand, hydrophobic interactions can be maintained even under the conditions used for SEM. Among all investigated glycols, NPG is the most hydrophobic molecule, and this characteristic can explain the blunt or unsharpened surface (Figure 7, Panel D). NPG and GLY share a more marked

propensity to three-dimensionality compared to linear DEG and TEG, noting that NPG is also the most hindered molecule. The larger molecular volume of NPG and relative three-dimensionality of GLY could hamper a uniform, well-sharpened coating. Nevertheless, the high hydrophobicity of NPG provided enzymatic stability even seven days after biosensor construction. The deposition of TEG (Panel E of Figure 7 and Panels 3A,3B of Figure S2 of the Supplementary material) resembles a tight overlapping of distinct pages, composed of an underlying and uniform PPD and the collapsed cratered network. In SEM images of the TEG biosensors, the glycol appeared tightly embedded in the matrix, making it potentially better able to stabilize the enzymes.

## 5. Conclusions

The stability of biosensor analytical parameters over time are essential features when developing biosensors for in vivo use. Based on the results obtained in the present study, glycols and TEG, in particular, can be used to improve biosensor performance. The results obtained here for glutamate biosensors could also be confirmed for other enzymatic biosensors, such as glucose and lactate biosensors. Studies of these biosensors are in progress already. Work is also ongoing to evaluate the ability of the containment net created by interacting glycols and proteins to protect the biosensors from extracellular enzymes, such as proteases, during in vivo applications.

**Supplementary Materials:** The following are available online at http://www.mdpi.com/2227-9040/8/2/23/s1, Figure S1: Calibration plots of two representative biosensors' designs in a 21 days period of time Ptc/PPD/PEI(1%)2/GlutOx5/DEG(0.1%) (Panel A); Ptc/PPD/PEI(1%)2/GlutOx5/TEG(0.1%) (Panel B), Figure S2: SEM of different biosensors' designs at 5000X (Panels A) and 10000X (Panels B) magnification. 1: Ptc/PPD/PEI(1%)2/GlutOx5/DEG(0.1%); 2: Ptc/PPD/PEI(1%)2/GlutOx5/NPG(0.1%); 3: Ptc/PPD/PEI(1%)2/GlutOx5/TEG(0.1%); 4: Ptc/PPD/PEI(1%)2/GlutOx5/GLY(0.1%).

**Author Contributions:** Conceptualization, G.R.; Methodology, G.R. and G.D.; Validation, G.R. and P.A.S.; Formal Analysis, P.M.; Investigation, A.B., P.A.; Writing—Original Draft Preparation, G.R.; Writing—Review & Editing, G.R. And P.A.S.; Visualization, S.M.; Supervision, G.R.; Project Administration, G.R.; Funding Acquisition, G.R. All authors have read and agreed to the published version of the manuscript.

**Funding:** Fondo di Ateneo per la ricerca 2019.

**Aknowledgments:** We sincerely thank Kusakabe (Yamasa Corp., Japan) for the gift of glutamate oxidase.

**Conflicts of Interest:** The authors declare no conflict of interest.

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
