# Peer review of "A New Perspective on Using Glycols in Glutamate Biosensor Design: From Stabilizing Agents to a New Containment Net"

_chemosensors, doi:10.3390/chemosensors8020023_

Round 1

Reviewer 1 Report

The authors have revised their manuscript "A new perspective on using glycols in glutamate biosensor design: from stabilizing agents to a new containment". This has improved the quality significantly. My recommendation is to accept the manuscript after some minor revision. 

Here are some comments and suggestions for the authors. I apologize if the authors already provided answer to some of my comments on their original manuscript. For some reason I was not able to access the (possible) letter they provided nor my original review. 

Line 66

There are some indications that (basal) glutamate concentrations might actually be in nanomolar range instead of micromolar owing to the effective transport systems.

Herman, M. & Jahr, C. (2007), ‘Extracellular glutamate concentration in hippocampal
slice’, Journal of Neuroscience 27(36), 9736–9741.

Vasylieva, N., Barnych, B., Meiller, A., Maucler, C., Pollegioni, L., Lin, J.-S., Barbier,
D. & Marinesco, S. (2011), ‘Covalent enzyme immobilization by poly(ethylene glycol)
diglycidyl ether (pegde) for microelectrode biosensor preparation’, Biosensors and
Bioelectronics 26(10), 3993–4000.

Line 79 (Equation 3)

If the reader is familiar with biosensors based on oxidase enzymes, this quite clear. However, it might be good to have a note that this reaction is not related to the enzyme but occurs on the electrode surface. 

Line 80

It is very bolt to state that H2O2 can be measured easily. I would say that H2O2 measurements on Pt surface at this high potentials are definitely not that easy, even though they are routinely used in several biosensor designs. In particular, as the authors have already stated in the manuscript, the issues with interfering species (AA, uric acid, dopamine + other cathecolamines) occurs at this potential and one needs to be very careful with their exclusion layers. Moreover, when it comes to biosensors requiring O2 for the enzyme, this high potential is pretty tricky due to Pt oxidation.

Line 130

Why were there four working electrodes? Were these simultaneously in the electrichemical cell? Also, why was stainless steel chosen as counter electrode?

Line 152

I still find the subscript numbers very confusing as they are not clearly described in the text. Moreover, as there were no other numbers of dips for GluOx for example, I don't think it is necessary to include the number here since it mostly causes confusion.

Figure 2

Data for concentrations below 25 uM is very unclear. It would be good to have a separate graph or another inset with these lower concentrations as the authors have stated that the relevant range for glutamate is 1-10 uM.

Line 173

Measuring glutamate at 50 mM seems a bit off as this is way higher than would be physiologically relevant. Why was so high concentration measured? 

Figure 3

Check the placement of 14 days and 21 days. Now they are confusing.

Line 266 (and elsewhere)

LRS values should be presented using current density instead of current to make it more comparable to other studies.

Table 1

Why is LOD suddenly presented in a table and not as a barplot as the other results? Check also placement of table caption.

Line 353

Remove the question mark. 

Line 358

Any thoughts or suggestions why TEG behaved differently from others? Some TEG properties are discussed later in the manuscript but maybe here some of those could be referenced here as well?

Line 419

Why is this data still omitted? AA was not mentioned in the results section and now it is suddenly brought up here in discussion. I would rather see these results compared to the SEM images as I don't think they add significant value to the manuscript. If you don't want to add AA data for some reason, please remove the part related to it from materials and methods as now it seems like it was really measured and that it is a part of the study.

Author Response

The authors have revised their manuscript "A new perspective on using glycols in glutamate biosensor design: from stabilizing agents to a new containment". This has improved the quality significantly. My recommendation is to accept the manuscript after some minor revision. 

Here are some comments and suggestions for the authors. I apologize if the authors already provided answer to some of my comments on their original manuscript. For some reason I was not able to access the (possible) letter they provided nor my original review. 

Line 66

There are some indications that (basal) glutamate concentrations might actually be in nanomolar range instead of micromolar owing to the effective transport systems.

Herman, M. & Jahr, C. (2007), ‘Extracellular glutamate concentration in hippocampal
slice’, Journal of Neuroscience 27(36), 9736–9741.

Vasylieva, N., Barnych, B., Meiller, A., Maucler, C., Pollegioni, L., Lin, J.-S., Barbier,
D. & Marinesco, S. (2011), ‘Covalent enzyme immobilization by poly(ethylene glycol)
diglycidyl ether (pegde) for microelectrode biosensor preparation’, Biosensors and
Bioelectronics 26(10), 3993–4000.

 At the suggestion of the referee, the part requested has been added in the introduction and the bibliographic references cited and added in the references paragraph.

Line 79 (Equation 3)

If the reader is familiar with biosensors based on oxidase enzymes, this quite clear. However, it might be good to have a note that this reaction is not related to the enzyme but occurs on the electrode surface. 

In agreement with the referee observation, the indication of the fact that the H2O2 oxidation occurs on the transducer surface has been highlighted by indicating the type of transducer in the reaction 3

Line 80

It is very bolt to state that H2O2 can be measured easily. I would say that H2O2 measurements on Pt surface at this high potentials are definitely not that easy, even though they are routinely used in several biosensor designs. In particular, as the authors have already stated in the manuscript, the issues with interfering species (AA, uric acid, dopamine + other cathecolamines) occurs at this potential and one needs to be very careful with their exclusion layers. Moreover, when it comes to biosensors requiring O2 for the enzyme, this high potential is pretty tricky due to Pt oxidation.

To meet the referee's observation, the term “easily” has been omitted from the sentence in the line 78

Line 130

Why were there four working electrodes? Were these simultaneously in the electrichemical cell? Also, why was stainless steel chosen as counter electrode?

Authors used 4 biosensors in order to have at the same time a number of biosensors sufficient for an adequate statistical analysis.

Stainless steel has been used as counter electrode because it is commonly used in three-electrode electrochemical cells (Vargas E et al Beverages 2017, 3, 22; Rothwell et al Electrochemistry Communications 10 (2008) 1078–1081; Twomey K Sensors 2006,6, 1679-1696). Moreover, the surface of the stainless-steel auxiliary electrode it is far wider than that of the working electrodes. This aspect become fundamental when high oxidative currents are generated.

Line 152

I still find the subscript numbers very confusing as they are not clearly described in the text. Moreover, as there were no other numbers of dips for GluOx for example, I don't think it is necessary to include the number here since it mostly causes confusion.

We appreciate the referee observation. Actually, we already answered in the previous submission to this question. That it is practice specifying the number of dips in the design of the biosensor, even they are of the same type (see Ford R, Devereux SJ, Quinn SJ, O'Neill RD Analyst. 2019 Aug 16;144(17):5299-5307.)

Authors would also to point out that the material and method section 2.3. Biosensor Construction and Characterization has been slightly modified in order to improve the clarity of the biosenosr construction procedures.

Figure 2

Data for concentrations below 25 uM is very unclear. It would be good to have a separate graph or another inset with these lower concentrations as the authors have stated that the relevant range for glutamate is 1-10 uM.

 In acceptance of the referee observation, Figure 2 has been modified in order to highlight the biosensor behavior in the 0-10 uM range of glutamate concentration. The legend of the Figure 2 has been modified in accordance.

Line 173

Measuring glutamate at 50 mM seems a bit off as this is way higher than would be physiologically relevant. Why was so high concentration measured? 

Authors reached up to 50 mM of glutamate concentration because they needed to reach the enzyme saturation in order to allow the statistical program to perform a correct non-linear regression (Michaelis-Menten equation) on calibration data, in order to extrapolate VMAX and KM parameters.

As known, VMAX represents the number of active molecules present on the transducer surface and it is obtained only when biosensor is exposed to high glutamate concentrations, that is when all the enzyme molecules are saturated. The higher the VMAX the higher the probability of interaction with glutamate-enzyme even at the lowest concentrations, producing hydrogen peroxide. Moreover, the Michaelis constant allows to define the affinity between the enzyme and the substrate, so this parameter becomes very important mainly at the lowest concentrations of glutamate.

 Figure 3

Check the placement of 14 days and 21 days. Now they are confusing.

We thank the referee for the observation. Actually, in all the figure the mistake has been changed.

Line 266 (and elsewhere)

LRS values should be presented using current density instead of current to make it more comparable to other studies.

 In acceptance of the referee's request, the LRS data were expressed as current density. In this regard, paragraph 2.4. Statistical Analysis has been modified as follows: “While VMAX was expressed as nA, LRS was expressed as current density (nA μM-1 cm-2).”

In addition, the paragraph 3.3. Impact of glycols on analytical parameters over time has been modified with the new data.

Table 1

In

 As requested by the referee, LOD table has been converted into bar plot, thus becoming Figure 6. The legend of the figure has been added and the numbering of the subsequent figures modified. All the variations have been highlighted.

Line 353

Remove the question mark. 

Thanks for the remark, typo has been corrected.

Line 358

Any thoughts or suggestions why TEG behaved differently from others? Some TEG properties are discussed later in the manuscript but maybe here some of those could be referenced here as well?

 We have already tried to explain the different behavior of TEG in the last part of Discussion paragraph:

“Surprisingly, the deposition of TEG revealed very interesting results both in terms of enzymatic kinetics (VMAX and KM) and analytical efficacy (LRS and LOD), over time. The high number of active molecules and high sensitivity, as well as the constant enzyme affinity, reflect the elevated capability of TEG to stabilize enzymatic activity, over the time frame studied here. Moreover, the persistence of many active enzyme molecules that were well-oriented for interaction with the substrate, could be due to hydrophilic and hydrophobic interactions of TEG through OH moieties and ether oxygens. Moreover, the structural flexibility of TEG probably allows better interactions with the amino acids of the enzyme. The TEG properties are evident in the SEM results (Figure 7, Panel E), which showed a more uniform layer in comparison with the other glycols. In SEM images of the TEG biosensors, the glycol appeared tightly embedded in the matrix making it potentially better able to stabilize the enzymes. Among all investigated glycols, NPG is the most hydrophobic and hindered molecule, and the latter characteristic can affect the surface of the biosensor (Figure 7, Panel D) since a larger molecular volume hampers a uniform coating. Nevertheless, the high hydrophobicity of NPG provided enzymatic stability even 7 days after biosensor construction.”

Line 419

Why is this data still omitted? AA was not mentioned in the results section and now it is suddenly brought up here in discussion. I would rather see these results compared to the SEM images as I don't think they add significant value to the manuscript. If you don't want to add AA data for some reason, please remove the part related to it from materials and methods as now it seems like it was really measured and that it is a part of the study.

In agreement with the referee observation, as AA measurements wasn’t an important focus of the present study, AA data have been omitted from the entire manuscript.

Reviewer 2 Report

In this paper the authors studied glycols as stabilizers for glutamate biosensors. The authors used diethylene glycol (DEG), neopentyl glycol (NPG), triethylene glycol (TEG) and glycerol (GLY) and evaluated different enzymatic and analytical parameters. The results show that the glycols studied, particularly TEG, improved the performance of the biosensors. The biosensors proposed by the authors showed good results and novelty. The work is well presented and easy to follow. In my opinion, the manuscript could be considered after minor revision.

In section 3.2, page 8, line 242. Please check the grammar of the sentence “…of the absence of glycol as final layer OR OF the presence of different glycols…”

In section 3.2, page 8, line 245. Please check the grammar of the sentence “In all designs, AND A general increase in…”

In the caption of Figure 5 a space is missing. Please, replace “Figure5” for “Figure 5”. The same was observed in page 15, line 449 with “Figure6”

In section 4, page 12, line 353 there is an exclamation sign. Please, check the sentence.  

Author Response

In this paper the authors studied glycols as stabilizers for glutamate biosensors. The authors used diethylene glycol (DEG), neopentyl glycol (NPG), triethylene glycol (TEG) and glycerol (GLY) and evaluated different enzymatic and analytical parameters. The results show that the glycols studied, particularly TEG, improved the performance of the biosensors. The biosensors proposed by the authors showed good results and novelty. The work is well presented and easy to follow. In my opinion, the manuscript could be considered after minor revision.

In section 3.2, page 8, line 242. Please check the grammar of the sentence “…of the absence of glycol as final layer OR OF the presence of different glycols…”

Authors thank the referee for the observation. Actually, the sentence wasn’t easy to follow so it has been amended as follows: “Figure 4 shows the effect of the presence of the absence of glycol as final layer on the KM over a 21-day period.”

In section 3.2, page 8, line 245. Please check the grammar of the sentence “In all designs, AND A general increase in…”

In the caption of Figure 5 a space is missing. Please, replace “Figure5” for “Figure 5”. The same was observed in page 15, line 449 with “Figure6”

In section 4, page 12, line 353 there is an exclamation sign. Please, check the sentence.  

Authors apologize for all the typos, which have been corrected.

Reviewer 3 Report

Article: A new perspective on using in glutamate biosensor design: from stabilizing agents to a new containment net.

Reviewer: The work written by Bacciu and collaborators describes the studies of glycol polymers for stabilization and immobilization through a containment network of enzymes for glutamate biosensors, evaluating how they influence their activity, using the parameters Vmax, Km, LRS, and LODs.

  1. From line 44 to 60 it is possible for the writing to be more concise for better fluidity of the text;
  2. The font size differs throughout the text, straighten;
  3. In line 152 and in other parts of the text, arrange the order of the biosensor layers, the most exposed is on the left;
  4. Figure 2 says it is representative, however, in the text below it is called and describes what was done in the work, so it should be in the results;
  5. During the materials and methods and the rest of the text, it is not clear which technique is used in assays this work;
  6. In figures 3, 4 and 5, days 14 and 21 are inverted, straighten;
  7. The results from line 212 to 230 can be rewritten to make the text more fluid;
  8. It is not described how the results of Vmax, Km, LRS and LODs were calculated;
  9. The SEM images could be enlarged for better viewing;
  10. In the Discussion, from line 426 to 448, it could fit together with sections 393 to 418, for a better understanding of the work and fluidity of the text;
  11. It is said in the paragraph that begins in line 419, about the AA study; however, the results of the same were not presented, it would be interesting to show them;
  12. The Discussion of SEM images could be much more explored.

Author Response

Article: A new perspective on using in glutamate biosensor design: from stabilizing agents to a new containment net.

Reviewer: The work written by Bacciu and collaborators describes the studies of glycol polymers for stabilization and immobilization through a containment network of enzymes for glutamate biosensors, evaluating how they influence their activity, using the parameters Vmax, Km, LRS, and LODs.

  1. From line 44 to 60 it is possible for the writing to be more concise for better fluidity of the text;

Although the manuscript has been revised by a mother tongue expert in the field, in acceptance with the referee's request, the part of the introduction indicated has been modified as follows:

As previously reported [16], these techniques are particularly suitable for patients, as well as animal models, for their non-invasiveness, although they are characterized by low spatial and temporal resolution, as well as a moderately high limit of detection [16].

In some studies, glutamate concentrations were measured in blood samples using High Performance Liquid Chromatography (HPLC) [17,18]. While blood sampling is more invasive than imaging, the approach is still suitable for use in patients and HPLC is reliable and sensitive enough to measure glutamate concentrations in blood samples, as well as in brain samples [17,19,20].

Given the pathophysiological role glutamate in the CNS and given that the etiology of some glutamate-related diseases is poorly understood, detection of glutamate in pre-clinical studies, in particular in animal models, has become increasingly important.

Since the early 1990s, microdialysis, a technique where a probe is inserted in tissue, has been the most widely used technique for monitoring neurochemicals in the CNS, including glutamate. Moreover, the coupling with HPLC allowed the measurements of numerous compounds present in microsdialysates obtained from extracellular fluids [21–23]. Low temporal resolution, the relatively large probe size (200 μm), as well as the need for coupling with an analytical technique to quantify analytes in the microdialysate are the most important limitations [16,24].

  1. The font size differs throughout the text, straighten;

Authors would thank the referee for the valuable observation. The font size has been made homogeneous

  1. In line 152 and in other parts of the text, arrange the order of the biosensor layers, the most exposed is on the left.

The authors would point out to the referee that the order of the layers that make up the biosensor design are in the correct order, as they are an exemplification of their construction protocol.

  1.  Figure 2 says it is representative, however, in the text below it is called and describes what was done in the work, so it should be in the results;

Authors thanks the referee for the observation. Figure 2 has been moved at the beginning of the Results paragraph, as it only wanted to be explanatory of the type of statistical analysis adopted on the data obtained from the calibrations.

  1. During the materials and methods and the rest of the text, it is not clear which technique is used in assays this work;

In agreement with the referee observation, at the end of paragraph 2.3 the following sentence “All the electrochemical procedures as electropolymerization or calibrations were carried out by means of constant potential amperometry (CPA)” has been added, in order to define which analytical technique has been used for all electrochemical procedures.

  1. In figures 3, 4 and 5, days 14 and 21 are inverted, straighten;

Authors would thank the referee for the important observation. Actually, the mistake has been corrected in all the figures.

  1. The results from line 212 to 230 can be rewritten to make the text more fluid;

In an attempt to further improve the fluidity of the text, this part of the results has been modified as follows:” Figure 3 shows the effect of the absence of glycol as final layer or of the presence of different glycols on the VMAX over a 21-day period. The control design at Day 1 showed a VMAX equal to 148.60 ± 3.48 nA, showing a significant decrease Vmax starting from Day 7 (126.60 ± 3.18 nA) and further decreasing up to Day 21 (13.36 ± 1.37 nA).The presence of DEG displayed a VMAX of 175.30 ± 3.78 nA on Day 1, while a small but significant decrease (p<0.001 vs Day 1) was observed on Day 7 (163.50 ± 3.27 nA) with further decreases on Day 14 and 21, dropping to 67.71 ± 2.45 nA.A comparable trend was observed when NPG was used. A reasonably high Vmax was obtained on Day 1 (253.20 ± 4.53 nA), but a significant decrease (p<0.001 vs Day 1) was observed over time, reaching a minimum value of 99.16 ± 4.53 nA at Day 21TEG biosensors produced the highest values of VMAX between Day 1 and 7 when the parameter was equal to 317.10 ± 5.46 nA (not statistically different from Day 1). Over time, VMAX decreased moderately but significantly (p<0.001 vs Day 1) on both Days 14 and 21, to a minimum of 221.50 ± 4.98 nA.GLY-loaded biosensors generated a VMAX of 222.20 ± 6.21 nA at Day 1 and a constant significant decrease (p<0.001 vs Day 1) was observed over time to 142.50 ± 5.22 nA at Day 21.”

  1. It is not described how the results of Vmax, Km, LRS and LODs were calculated;

The authors would to point out to the referee that in paragraph 2.4. Statistical Analysis the modalities, with which all the parameters considered in the work were calculated, has been stated

  1. The SEM images could be enlarged for better viewing;

In apcettance of the referee remark, the SEM images have been enlarged in order to improve their viewing.

  1. In the Discussion, from line 426 to 448, it could fit together with sections 393 to 418, for a better understanding of the work and fluidity of the text;

As requested by the referee, both parts have been connected, in order to improve comprehensibility and fluidity of the text.

  1. It is said in the paragraph that begins in line 419, about the AA study; however, the results of the same were not presented, it would be interesting to show them;

In accordance with another referee, the part regarding AA results has been omitted as it wasn’t the main focus of the present project and did not bring any further improvements to the results

  1. The Discussion of SEM images could be much more explored.

As requested by the referee comment on SEM images have been improved and added along the 3.4. Scanning Electron Microscopy (SEM) Analysis and the Discussion paragraph.

Round 2

Reviewer 1 Report

The authors have revised their manuscript and provided answers to my previous comments. Thus, I would like to recommend the publication of the manuscript. 

Reviewer 3 Report

No comments

This manuscript is a resubmission of an earlier submission. The following is a list of the peer review reports and author responses from that submission.

Round 1

Reviewer 1 Report

The grammar of the manuscript needs to be reviewed and edited to improve the flow of the text make the entire work easier to understand.  Include another panel in figure 2 to show the 1 mm x 125 um tip of the electrode as a whole, this will help explain the geometry Figure 2 has a very low resolution and does not demonstrate the organization of your sensor well Why did you use Pt/Ir instead of 99.9% Pt? Can you account for any residue from the coating interfering with your surface modification?  Your abbreviations in the text are inconsistent this makes it confusing!   How are you estimating 'layers' of molecules like glycol? Does it have a specific thickness? How stable is that coating as a non-specific adsorption? "Dependently on the glycol, different biosensors were manufactured with the following general design:" What does this statement mean? it's not very clear. Did you determine the thickness of your enzyme layer or what distance the hydrogen peroxide will need to diffuse in order to react with the electrode surface? Is it necessary to expose your enzyme to a polarizing potential overnight? or to pure DI water? Prolonged exposure to these conditions tend to have a negative effect on protein conformation. If you cannot gain a stable signal without normalizing overnight you may have a problem with your equipment  For figure 4 are you able to convert this data from nA to [Glut]/time. You should easily be able to convert this to # Glut oxidized per second. The rate of glutamate detection is what is really telling us something about the sensitivity of the sensor. How did you account for electrical drift that may occur between measurements on different days? Was there some kind of base line subtraction or zeroing?  For figure 4 what is your n value and what statistical test did you use for significance? Without this information your statements about significance don't mean anything. How do your Km measurements on the sensor compare to the enzyme in solution? Does your coating have any effect on this parameter? Again how are you assessing significance? What test are you using what is the value of n? You haven't defined what LSR is or how your measuring it. It looks like it is just the slope of the calibration plot. Why not include the calibration plots? Then you can also provide a limit of detection which is a vital parameter for these sensors.  For figure 7 were you able to generate XPS data at the same time? The images as they are don't show much about the architecture of the sensor and don't show a lot of detail. What is causing the heterogeneity on the surface of some of the sensors? is it an incomplete coating or an imperfection? You can't argue that the 'craters' are due to a certain substance without further chemical evidence.  Does your coatings make the sensor harder to foul? Did you look at possible contaminating substances? With no limit of detection, no determination of if other substances will generate signals, or examination of how the sensor will survive in a biological environment your manuscript is missing some key elements necessary to understand how your sensor compares to other published sensors or even how one coating may truly compare to another.  Your particular coating glycols are not covalently attached. This may be an asset for your sensor, but have you determined that they remain associated with the surface over time? Might they dissociate, be removed in bulk, or become rearranged over the course of your measurements and storage conditions? You really need to measure and discuss points like this to have a complete understanding of the benefits of each of the molecules you tested.  

While this manuscript does have some novelty in the use of small molecules as an outer coating on the biosensor surface I do not recommend publishing at this point. I suggest the authors add additional techniques and methodologies and improve their statistical analysis and the level of scientific writing in the manuscript before it is reconsidered. 

Reviewer 2 Report

The authors present in their paper the use of four different glycols for improving the stability of glutamate biosensors. The key parameters they use in their evaluation are Vmax, KM and the slope in the linear region. The topic is interesting and the results show some promise but there are major issues with the manuscript. The general feeling is that the manuscript is somewhat unfinished and requires more work in regard of experiments and analysis before publication. 

I provide here some comments and suggestions that hopefully help the authors to improve the manuscript for resubmission. 

Page 3, line 112

The authors argue that the linear region slope is the most important analytical parameter for biosensors. This is only partially true as especially for several neurotransmitters the ability to measure very small concentrations and fast transients are equally - or even more - important.  The authors should define in the manuscript at least the LODs for their sensors as well as the smallest measured amount of glutamate. 

2. Materials and Methods

Page 4, line 130

Was the enzyme received in solution or as a powder? In the latter case, it should be mentioned in which solvent was it dissolved in. Late (on line 154) it is said that electrodes were dipped in GluOx solution but it is unclear what was the concentration.

Page 4, line 132

Why did the authors use sodium glutamate instead of glutamic acid?

Page 4, line 137

No software is mentioned in this section.

Page 4, line 140

The concentration of Ag/AgCl reference electrode filling solution is mentioned but not what it is. 

Page 4, line 148

It is not clearly defined in the text what the abbreviation PPD means. It is in the figure captions but it must be written also in the text. There are also some other abbreviations that are not clearly defined (LRS in the abstract, HPLC, ECF). On the other hand, there are also unnecessary abbreviations such as HP for H2O2 or Glut for glutamate.  

Page 4, line 159

It is not relevant information to have the number of dips as subscripts in the electrode names since all of the different types were prepared the same way. 

Page 5, figure 2

This figure does not add much to the paper as such. Showing a general structure for the sensors and combining that with figure 1 would be more meaningful. 

Page 5, figure 3

The interesting region for glutamate sensors is below 1 mM. I would prefer that the inset was the actual figure and the MM data fitting would be the inset. Or that both were larger (A and B parts of figure 3).

Page 6, line 174

It is not defined here clearly what does "a full glutamate calibration" mean. Moreover, it should be explained more clearly what kind of experiments were performed. Now it is only stated that electrodes were held at 0.7 V vs. Ag/AgCl overnight for stabilization. In addition, the procedure for AA calibration should be similarly explained. 

Page 6, line 185

Here it is said that LRS stands for linear regression even though it has been previously defined as linear region slope. 

Page 6, line 189

Were the sensors inspected also after the experiments? Were there some changes in the coatings?

Page 6, figure 4

Green and red bars next to each other, not good for color blind persons. Maybe authors could consider using different textures instead of colors? Moreover, the numbers for different days look somewhat messy. 

Pages 7-8, figures 4 and 5

The statistical differences between days for individual electrodes are shown in the figures and discussed in the manuscript. It would be interesting to know if there were statistical differences between different electrodes? Now the differences are mainly compared based on magnitudes of the parameters.

Page 8, line 249

Current arising from electrochemical reactions is proportionate to the sensor area. Here (and in the whole manuscript) instead of current values, current density would be more meaningful in terms of comparing the obtained results for literature values. It would be also important to have comparison to literature to make some conclusions about the sensor performance. 

In addition, there are some error values given here to the LRSs. How were these calculated? If these are standard deviations, how many electrodes per type were used in the study?

Page 9, figure 7

The black bars should be removed and clearer scale bars added to the micrographs. 

Page 10, line 313

Glutaraldehyde is mentioned here as an aggressive crosslinker that might affect enzyme performance. The authors should have included also it and/or for example Nafion for comparison. Or maybe a sensor without any additional coating (only PEI + GluOx). Now the study is missing a clear point of reference. 

Page 11, line 375

It is not clearly evident from the SEM micrographs that TEG structure really explains the better enzyme stability. It would be more interesting to see for example a cross-section of the coatings than the surface to see if the internal structure and the parts that are in contact with the enzyme are somehow different.

Finally, could it be that instead of enzyme stability, the storage affects the glycols' ability to let glutamate through?

General notes

The authors performed some experiments with ascorbic acid but the results are not presented or even commented in the manuscript.  It would give the manuscript more credibility if enzyme activity was measured also with a fluorometric enzyme assay kit and the MM fits compared.  The same data is used for defining LRS, KM and Vmax and it is even stated in the manuscript that LRS is a function of the other two. Is this the case for the data presented in the manuscript? It would be interesting to see if this applies for the results obtained here.  The sensor performance is not compared to literature.  There is a supplementary consisting of additional SEM micrographs. This is not referenced in the manuscript and it should be removed as it does not bring any additional value to the manuscript. 

Reviewer 3 Report

Addition of key finding result in the abstract need to be added to captures the reader’s interest and imagination. Addition of a few illustrations especially where the applications are concerned to (in very specific way) capture the readers interest and imaginations, in the introduction section. Future trends and prospects need to be elaborated further, stipulating the research directions. Under experimental section, author need to be describe all the used chemicals with their manufacturer names, purity and place of manufacturing materials under material section. Error bars are missing for the associated data. Author need to show the reproducibility for his result not just randomly. Honestly, author need to response, how they find the sensing response for the glutamate sensor as compared to standard glutamate sensor. What is the market value of this type of sensor. Quality of all the presented figurer in manuscript is VERY POOR, author need to improve the quality of the figures before publication. It is highly recommended that author need to use MATLAB or Origin software draw the figure not the Excel. Why using the drop casting methods for the modifying the surface of graphite electrode for the MWCNT, while other approach will more surface will be more appropriate for surface modification. What is the active surface area of the different type of graphite sheet? It is very crucial report and it need to know the information about active surface area. What is the future prospect of this study in comparison to the already reported manuscript about glutamate biosensor?